# The Profile of MicroRNA Expression and Potential Role in the Regulation of Drug-Resistant Genes in Doxorubicin and Topotecan Resistant Ovarian Cancer Cell Lines

**DOI:** 10.3390/ijms23105846

**Published:** 2022-05-23

**Authors:** Piotr Stasiak, Dominika Kaźmierczak, Karol Jopek, Michał Nowicki, Marcin Rucinski, Radosław Januchowski

**Affiliations:** 1Department of Anatomy and Histology, Collegium Medicum, University of Zielona Gora, Zyty 28 St., 65-046 Zielona Gora, Poland; r.januchowski@cm.uz.zgora.pl; 2Department of Histology and Embryology, Poznan University of Medical Sciences, 61-781 Poznan, Poland; dominika.ka.poznan@gmail.com (D.K.); karoljopek@ump.edu.pl (K.J.); mnowicki@ump.edu.pl (M.N.); marcinruc@ump.edu.pl (M.R.)

**Keywords:** ovarian cancer, second line chemotherapy, microRNA, drug-resistant genes

## Abstract

Epithelial ovarian cancer has the highest mortality among all gynecological malignancies. The main reasons for high mortality are late diagnosis and development of resistance to chemotherapy. Resistance to chemotherapeutic drugs can result from altered expression of drug-resistance genes regulated by miRNA. The main goal of our study was to detect differences in miRNA expression levels in two doxorubicin (DOX)- and two topotecan (TOP)-resistant variants of the A2780 drug-sensitive ovarian cancer cell line by miRNA microarray. The next aim was to recognize miRNAs as factors responsible for the regulation of drug-resistance genes. We observed altered expression of 28 miRNA that may be related to drug resistance. The upregulation of miR-125b-5p and miR-935 and downregulation of miR-218-5p was observed in both DOX-resistant cell lines. In both TOP-resistant cell lines, we noted the overexpression of miR-99a-5p, miR-100-5p, miR-125b-5p, and miR-125b-2-3p and decreased expression of miR-551b-3p, miR-551b-5p, and miR-383-5p. Analysis of the targets suggested that expression of important drug-resistant genes such as the collagen type I alpha 2 chain (*COL1A2*), protein Tyrosine Phosphatase Receptor Type K (*PTPRK*), receptor tyrosine kinase—*EPHA7*, Roundabout Guidance Receptor 2 (*ROBO2*), myristoylated alanine-rich C-kinase substrate (*MARCK*), and the ATP-binding cassette subfamily G member 2 (*ABCG2*) can be regulated by miRNA.

## 1. Introduction

Epithelial ovarian cancer (EOC) is a heterogeneous malignancy with variable clinical development, and it has the highest mortality among gynecological cancers [1]. GLOBOCAN data from 2018 indicated 295,414 new EOC cases and 184,799 ovarian cancer-related cases of mortality [2]. Late diagnosis, usually at stage III or IV according to FIGO, and development of drug resistance are the main reasons for the high mortality rate among ovarian cancer patients [3,4]. The standard treatment in ovarian cancer is surgical resection followed by chemotherapeutic treatment [3,5]. The first line is always composed of a combination of platinum derivatives and taxanes (paclitaxel—PAC) [5]. Only 5% of patients are primarily resistant to this treatment. The others develop drug resistance after different times of treatment. About 17% of the patients have recurrences within six months after treatment, and this group is designated as not sensitive. Approximately 23% of the patients are classified as partially sensitive to platinum, and they have a recurrence within 6–12 months after treatment. The patients that develop recurrence after 12 months or more are qualified as sensitive to platinum (roughly 55%). Although these patients are sensitive at the beginning of treatment, only 20% of the patients are believed to be cured (over 120 months without recurrence), and others develop progression within this time [5,6]. Thus, for most ovarian cancer patients, the second line of chemotherapy is required.

Doxorubicin (DOX) and topotecan (TOP) are some of the most important drugs that are used in the second line of chemotherapy for platinum-resistant patients [7,8]. TOP, a semisynthetic derivative of camptothecin [9], is used in chemotherapy of relapsed ovarian cancer [10] and small cell lung cancer (SCLC) [11], among others. It acts by the inhibition of DNA topoisomerase I [12]. Stabilization of the enzyme–DNA complex by TOP results in the inhibition of DNA replication and transcription, leading to cancer cell death [13]. However, cancer cells can develop different mechanisms of TOP resistance. The most important transporter responsible for the active removal of TOP from cancer cells is BCRP (Breast Cancer Resistant Protein) encoded by the *ABCG2* gene. Expression of this gene was reported among others in breast and ovarian cancer and cancer cell lines [14,15,16]. TOP resistance can also be related to the overexpression of the *ABCB1* (*MDR1*) gene, encoding the glycoprotein P (P-gp) [16,17,18]. DOX is a very important drug used in the therapy of many cancers, including the second line of ovarian cancer chemotherapy [19,20]. At a molecular level, DOX is an inhibitor of DNA topoisomerase II that acts mainly after replication, forming irreversible covalent cross-links between the topoisomerase and DNA, leading to DNA breakage and consequently to cell death [21]. Additionally, DOX, as a planar compound, intercalates into the DNA structure, inhibiting transcription and replication [22]. The main mechanism underlying the resistance to DOX is the expression of P-gp [23]. Overexpression of P-gp was also observed by our team in DOX-resistant ovarian cancer cell lines [16,18].

However, the expression of drug transporters is not the only mechanism of resistance to cytotoxic drugs. Recently, we observed changes of expression of many new drug-resistant genes that seem to be specific for DOX and TOP resistance [24]. Among others, we observed the overexpression of Interferon Gamma Inducible Protein 16 (IFI16) in DOX-resistant cell lines [25], as well as the overexpression of Sterile Alpha Motif Domain Containing 4A (*SAMD4*) in TOP-resistant cell lines [17]. In all investigated cell lines, we also observed the decreased expression of Tyrosine Phosphatase Receptor Type K (PTPRK) that seems to be related to increased intracellular signal transduction [26,27]. Drug resistance in cancer can also be related to the expression of extracellular matrix (ECM) molecules [28,29,30]. Recently, we reported the upregulation of many collagen genes, especially in TOP-resistant ovarian cancer cell lines [31,32]. Abundant expression of ECM molecules can limit drug penetration into the tumor tissue [33] as some cytotoxic drugs, such as DOX or vinblastine, can bind to cellular macromolecules, limiting their availability to the tumor cells [34]. ECM molecules can also induce cell adhesion-mediated drug resistance (CAM-DR) by binding to cellular receptors [28,35].

Drug resistance can be linked to the activity of many well-researched genes. However, the exact way in which their expression is regulated may require further investigation. One of the factors responsible for change in gene expression is microRNA (miRNA). These small molecules, only 19–29 nucleotides in length, take part in post-transcriptional regulation. First discovered by Victor Ambros’ team in 1993 during research on *Caenorhabditis elegans* [36], they are a vast group of single-stranded RNAs capable of ubiquitous modulation of protein expression by interaction with mRNA [37]. It is worth noting that a single miRNA can target multiple mRNAs of various genes responsible for cancerous processes, whereas a single mRNA can bind several different molecules of miRNA [38]. miRNA-encoding genes can be located both within introns and exons of other genes. They are classified in groups called families and are typically transcribed by RNA polymerase II in the nucleus [39]. Following transcription within the nucleus, an immature, hairpin-shaped pri-mRiNA is cleaved by an enzyme called Drosha and its cofactor DGCR8 into pre-miRNA that is secreted to the cytoplasm by exportin 5. There, it undergoes another cleavage that is conducted by the Dicer protein—and mature, single-stranded miRNA is created [40]. Then, it is incorporated into the RISC complex, capable of binding to mRNA thanks to an antisense seed sequence present within the microRNA [41]. The seed sequence is located in the 5′ region of the miRNA and is complementary to the 3′ untranslated region of mRNA [42]. If the complementarity between miRNA and mRNA is low, the translation process is downregulated, and less protein is produced, and if both strands have a strong match, mRNA becomes cleaved before the translation may occur [37]. So far, over 2000 miRNAs capable of regulating about one third of the human transcriptome have been predicted by bioinformatic analyses [42,43].

In healthy cells, miRNAs take part in homeostasis processes. Changes in their expression are often responsible for a plethora of diseases, including cancer [44]. In cancers, miRNAs may target different genes playing suppressor or oncogenic functions [45]. It is thought that some micro RNAs are involved in carcinogenesis by dysregulation of cell growth signals, evasion of apoptosis, changes in cell metabolism, and induction of angiogenesis and the epithelial–mesenchymal transition (EMT), resulting in increased tumor growth and metastasis formation [41]. In the case of EOC, some microRNA suppressors have already been identified, namely miR-145, miR-200c, and miR-192, together with some oncogenes—miR-572, miR-1207, and miR-551b [40]. Circulating miRNA in body fluids can reflect cancer tissue expression and are very stable, which makes circulating microRNA a promising biomarker in cancer diagnosis, prognosis, and a potential therapeutic agent in cancer treatment [46,47].

There is more and more evidence supporting the notion that non-coding RNA takes an important part in the development of drug resistance in cancer, including EOC [48,49]. When compared, drug-sensitive cell lines manifested different miRNA expression levels than their drug-resistant counterparts, and the same was observed in the case of the tissue samples [41]. According to our previous studies, in EOC cell lines resistant to therapeutic agents, 40 miRNAs are expressed at different levels than in drug-sensitive ovarian cancer cells [50]. In another study, we observed changes in 46 miRNAs that seem to be related to CIS and/or PAC resistance [51]. It was established that patients suffering from PAC-resistant EOC face better prognosis when their miR-663 and miR-622 levels are lower. A different study concluded that EOC cell lines resistant to PAC showed decreased expression of miR-31, and overexpression of this miRNA led to sensitization of ovarian cancer to PAC, both in in vitro and in vivo conditions [52]. Similarly, decreased expression of miR-31 was described by us in PAC-, CIS-, and TOP-resistant cell lines [50]. In our previous study, we also observed that miRNAs regulate the expression of key drug-resistant genes. Among others, we observed strongly downregulated miR-29a in TOP-resistant cell line. Its target—Collagen Type III Alpha 1 Chain (*COL3A1*), a gene responsible for TOP resistance—is upregulated in these cells [50]. Cell lines resistant to PAC exhibit miR-363 downregulation combined with upregulation of the *MDR1* gene [50]. In another PAC-resistant cell line, the expression of the *MDR1* gene was regulated by miR-21-5p [51]. Changes in miRNAs expression also correlated with increased expression of receptor protein-tyrosine kinases *EPHA7* and decreased expression of protein phosphatase *PTPRK* [51]. In both of our studies, downregulation of tumor suppressor gene *SEMA3A* in PAC-resistant cell lines correlated with the upregulation of miR-145 [50,51].

The usage of miRNA microarrays for identifying specific microRNAs involved in cancer resistance has previously proven to be accurate and effective [51]. In our current study, it served to investigate changes in miRNA expression in DOX- (A2780DR1, A2780DR2) and TOP-resistant (A2780TR1, A2780TR2) ovarian cell lines. Some miRNAs exhibited altered expression levels, with some being downregulated and others upregulated. For part of the miRNA, we also identified targets among drug-resistant genes.

## 2. Results

### 2.1. Gene Chip Quality Assessment

In the present study, we used standard factors such as signal-to-noise ratio internal hybridization and controlled spike-in-controls to determine the quality of analyzed samples preliminarily. Controlled spike-in-controls were spike_in-control-2, spike_in-control-23, spike_in-control-29, spike_in-control-31, and spike_in-control-36. Oligos 2, 23, and 29 were RNA, confirming the poly(A) tailing and the ligation. Oligo 31 (poly(A) RNA) confirmed ligation. Oligo 36 was poly(day) DNA and confirmed ligation and the lack of RNases in the RNA sample.

### 2.2. Gene Expression Evaluation and Gene Expression Lists

In this study, we analyzed the miRNA transcriptional profiles of two DOX-resistant and two TOP-resistant ovarian cancer cell lines. The variations observed in our results shined new light on miRNA expression changes during drug-resistance development in ovarian cancer. When compared to the drug-sensitive A2780 cell line, miRNA expression changed significantly in the DOX- and TOP-resistant sublines, as shown in Table 1. In order to distinguish aberrantly expressed miRNAs from those with regular expression, we selected only miRNAs that were expressed at levels higher than 5-fold or lower than 0.2-fold (with up- or down-regulation of more than/less than 5 and −5, respectively) and a *p*-value < 0.05. The miRNA expression profiles of the drug-resistant cells were compared against their counterparts that exhibited drug-sensitivity.

### 2.3. miRNAs Expression in DOX and TOP Resistant Cell Lines

Figure 1 shows the general profile of miRNA expression in the experimental group. For 61 miRNAs, the changes in their expression were observed. Fifty-four miRNAs had upregulated expression in at least one drug-resistant cell line. Seven miRNAs exhibited downregulated expression in at least one drug-resistant cell line (Table 1). To further tighten up the analysis, in order to increase specificity to DOX- and TOP-resistant cells, only the following miRNAs were selected: (1) ones with miRNA expression changed at least five-fold (fold ±5, *p* < 0.05) in both cell lines resistant to the same drug (DOX or TOP respectively); (2) ones with at least 10-fold changes in expression of the miRNA in one drug-resistant cell line. As a consequence, we observed notable changes in expression of 28 miRNAs. The expression level of 23 miRNAs was upregulated in at least one drug-resistant cell line. Five miRNAs were downregulated in at least one cell line resistant to drug (Table 2, Figure 2).

The expression of two miRNAs was changed in all investigated cell lines, and among them, expression of miR-218-5p was downregulated, and expression of miR-125b-5p was upregulated. The expression of five miRNAs was changed in three of four cell lines, and among them, the expression of miR-99a-5p, miR-100-5p, and miR-125b-2-3p was upregulated in both TOP-resistant cell lines and in the A2780DR1 cell line. The expression of miR-551b-3p and miR-551b-5p was downregulated in both TOP-resistant cell lines and in the A2780DR1 cell line. Additionally, upregulation of miR-935 was observed in both DOX-resistant cell lines, and downregulation of miR-383-5p was observed in both TOP-resistant cell lines.

Thus, the upregulation of miR-125b-5p and miR-935, and the downregulation of miR-218-5p, was specific to DOX resistance. On the other hand, the upregulation of miR-99a-5p, miR-100-5p, miR-125b-5p, and miR-125b-2-3p, and the downregulation of miR-218-5p, miR-551b-3p, miR-551b-5p, and miR-383-5p were specific to TOP resistance. We also observed very high similarity between both TOP-resistant cell lines and the A2780DR1 cell line: overexpression of miR-99a-5p, miR-100-5p, and miR-125b-2-3p and downregulation of miR-551b-3p and miR-551b-5p were observed in these cell lines.

The expression of another 19 miRNAs was changed at least 10-fold in one resistant cell line. Among them, the expression of 18 miRNAs was upregulated, and the expression of one miRNA was downregulated. Between all analyzed miRNAs, changes in expression of nine miRNAs were very significant (>20 fold). An enormous increase in expression was observed in the case of miR-217, which was upregulated over 200-fold in the A2780DR1 cell line. Among other strongly-upregulated miRNAs, we could distinguish those upregulated over 40-fold (miR-205-5p in the A2780TR1 cell line, miR-216b-5p in the A2780DR1 cell line), and those upregulated over 30-fold (miR-216a-5p in the A2780DR1 cell line, and miR-379-5p in the A2780TR1 cell line). Four miRNAs were upregulated over 20-fold: miR-34a-5p in the A2780DR2 cell line, and miR-125b-5p A2780DR1 and both TOP-resistant cell lines, miR-127-3p and miR-487b-3p, in the A2780TR1 cell line. Among the strongly downregulated miRNAs we observed strong downregulation (over 40-fold) of miR-551b-3p in the A2780TR2 cell line. These miRNAs were also strongly downregulated in the A2780DR1 cell line (over 30-fold) and in the A2780TR1 cell line (over 20-fold). We also observed an over 20-fold downregulation of the let-7i-5p in A2780TR2 cell line.

### 2.4. Analysis of Target Genes Expression

The goal of the second part of our investigation was to determine if the described miRNAs are involved in the regulation of genes responsible for the development of drug resistance. Assuming that an increase in miRNA expression leads to a decrease of target gene expression and the other way around, for the next part we selected only those target genes that had their fold change value inversely correlated with miRNA fold change. We applied identical cut-off values for miRNAs and their targets—at a minimum 5-fold up/down-regulation and adjusted *p* value <0.05. Target expression below these cut-off criteria was regarded as “not significant (NS)” when the target gene lists were made. The microarray data used was published by us previously [24,53,54]. With these criteria (5-fold up- or downregulation change), the targets for 30 out of 61 miRNAs were found.

For further tests, we selected target genes related to drug resistance, the extracellular matrix, and cancer stem cell biology from the whole population of target genes with the use of the following key words in the Gene Ontology (GO) database: a stem cell, collagen-containing extracellular matrix, drug transport, extracellular matrix, extracellular space, response to a drug. In our previous works, we described that these genes belong to differentially expressed genes in cell lines with drug resistance [24,53,54].

In the A2780DR1 cell line, we identified targets for eight miRNAs (Figure 3). Among others, we observed the downregulation of two SLC (solute carrier) transporters: *SLC27A2* correlated with the upregulation of miR-216a-5p and miR-216b-5p, and *SLC6A15* correlated with the upregulation of miR-137. The upregulation of miR-137 correlated also with *PTRPD* (*Protein Tyrosine Phosphatase Receptor Type D*) downregulation. The increased expression of *CDH2* (*Cadherin 2, Type 1, N-Cadherin*) was associated with the downregulation of miR-218-5p.

In the second DOX-resistant cell line, A2780DR2, we identified targets for seven miRNAs (Figure 4). As drug-resistance development is associated with increased signal transduction, we observed the downregulation of phosphatase *PTPRK* (*Protein Tyrosine Phosphatase Receptor Type K*) by miR-935 and the upregulation of protein kinase *EPHA7* (*EPH Homology Kinase 3*) associated with the downregulation of miR-218-5p, among others.

Topotecan is a very important drug in the second line of ovarian cancer chemotherapy. Thus, we also analyzed the expression of miRNAs in two TOP-resistant cell lines. In the A2780TR1 cell line, targets for eight miRNAs were present (Figure 5). Among others, we observed the downregulation of protein phosphatases. The decreased expression of *PTPRK* correlated with the increased expression of miR-409-3p, and miR-431-5p upregulation correlated with the decreased expression of *PTPRZ1* (*Protein Tyrosine Phosphatase Receptor Type Z1*). The upregulation of *SAMD4* (*Sterile Alpha Motif Domain Containing 4A*) was associated with the decreased expression of miR-551b-5p. The downregulation of miR-551b-5p and miR-218-5p was associated with the increased expression of MARCKS (*Myristoylated Alanine-rich C-Kinase Substrate*). Here, we also observed that the increased expression of the most important TOP-resistant gene, *ABCG2* (*ATP Binding Cassette Subfamily G Member 2*), was associated with miR-212-3p downregulation.

In the second TOP-resistant cell line, A2780TR2, we identified targets for seven miRNAs (Figure 6). As in all the other resistant cell lines, we observed the downregulation of protein phosphatases *PTPRK* and *PTPRZ1*; however, in this case, it was caused by other miRNAs than in the A2780TR1 cell line. Here, the downregulation of *PTPRK* correlated with miR-363-3 upregulation, and the decreased expression of *PTPRZ1* was observed together with the increased expression of miR-20b-3p. The increased expression of miR-363-3p correlated with the downregulation of the *ROBO2* (*Roundabout Guidance Receptor 2*) gene. The upregulation of the *SAMD4* gene was associated with the downregulation of miR-551b-5p. As in the A2780TR1 cell line, the overexpression of *MARCKS* correlated with the downregulation of miR-551b-5p and miR-218-5p. The upregulation of *S100A10* (*S100 Calcium Binding Protein A10*) was regulated by miR-10b-3p. The increased expression of ECM components—*ITGB8* (*Integrin Subunit Beta 8*), *MMP1* (*Matrix Metallopeptidase 1*), *COL1A2* (*Collagen Type I Alpha 2 Chain*), and *COL15A1* (*Collagen Type XV Alpha 1 Chain*)—was associated with let-7i-5p downregulation.

## 3. Discussion

As most ovarian cancer patients develop drug resistance, it is a very important problem in the treatment of this type of cancer. Drug resistance is a complex phenomenon, as hundreds of genes play a role in different mechanisms of drugs resistance. Although we know many of them, the regulation of their expression is still poorly understood. In mammals, as much as 60% of genes can be regulated by miRNAs [37]. miRNAs are considered not only as regulators of gene expression but also as molecular markers of drug resistance and ovarian cancer progression [41]. Thus, discovering their role in the development of drug resistance seems to be crucial for understanding the whole process.

Here, we investigated the expression of micro RNA genes in ovarian cancer cell lines resistant to DOX and TOP—drugs used in the second line of ovarian cancer chemotherapy [7,8]. This paper is a continuation of our previous study describing the role of miRNA in the regulation of drug-resistance genes in ovarian cancer cell lines resistant to CIS and PAC—drugs commonly administered in the first line of ovarian cancer chemotherapy [51]. What should be mentioned is that all cell lines studied in the current and previous study were developed from the drug-sensitive A2780 cell line [16]. In contrast to many other studies where only one pair of drug-sensitive/resistant cell lines is tested, here we analyzed changes of miRNA expression in two DOX- and two TOP-resistant cell lines. It is a unique model for such investigation. Previously, we characterized these cell lines according to drug-resistance genes expression. We observed overexpression of the most important drug transporters: P-gp in both DOX-resistant cell lines, and BCRP in both TOP-resistant cell lines [16]. Additionally, we observed changes in expression of many genes encoding ECM molecules [54]. We also noted up- or downregulation of “new drug-resistance genes” that may be related to DOX and/or TOP resistance [17,24,55]. Both TOP-resistant cell lines also showed cross resistance to DOX [16].

To make our analysis more specific we started by identifying miRNAs that changed expression in both DOX- or both TOP-resistant cell lines, respectively. Among the 28 miRNAs analyzed in this experiment, the expression of two (miR-935 and miR-125b-5p) was upregulated and one (miR-218-5p) was downregulated in both DOX-resistant cell lines, suggesting their role in DOX resistance. We did not find any data concerning the role of miR-935 in DOX resistance; however, miR-935 overexpression seems to be associated with PAC resistance in non-small cell lung cancer (NSCLC) [56]. Increased expression of miR-935 was also observed in hepatocellular carcinoma (HCC), where it promoted cell proliferation and tumorigenesis [57], and in both clear cell renal cell carcinoma (ccRCC) [58] and colorectal cancer (CRC), where it promoted cell proliferation, migration, and invasion [59]. Thus, we observed increased expression of miR-935 in drug-resistance development and others during tumor progression, suggesting that it may be involved in both processes. Here we found increased expression of miR-125b-5p in both DOX- and in both TOP-resistant cell lines. Previously we also observed overexpression of miR-125b-5p in two CIS- and one PAC-resistant cell lines [51]. Thus, in seven out of eight investigated cell lines, we observed overexpression of miR-125b-5p. This suggests that overexpression of this miR can be an unspecific marker of drug resistance in our A2780 drug-resistance development model. Looking through the literature, we did not find any data regarding a correlation between miR-125b-5p expression and DOX or TOP resistance. In the EOC study, the miR-125b-5p serum level was decreased in comparison to the control. Furthermore, it was lower in chemoresistant patients than in chemosensitive ones and lower in advanced cancer (FIGO stage III + IV), and a low level of miR-125b correlated with poor prognosis [60]. Thus, our result seems to contrast other observations made in ovarian cancer. The expression of miR-218-5p in both DOX- and both TOP-resistant cell lines was downregulated. Previously, we also observed the downregulation of this miRNA in two CIS- and two PAC-resistant cell lines [51]. Taken together, the expression of miR-218-5p was downregulated in all cell lines developed from the A2780 cell line, suggesting its important role in drug-resistance development. Our results seem to be in accordance with the data obtained from ovarian cancer patients. In a study by Wang et al., a lower expression level of miR-218-5p was observed in ovarian cancer cell lines in comparison to normal cell lines, and in tissue from ovarian cancer in comparison to normal ovary. The level of miR-218-5p was also inversely correlated with tumor progression and with FIGO—it was decreased in stage III/IV and in patients with metastasis [61]. Because metastases are often more resistant to chemotherapy and we observed downregulation of miR-218-5p in all drug-resistance cell lines, we suppose that downregulation of miR-218-5p can be associated with both processes.

Many more similarities were observed between both TOP-resistant cell lines. Except for upregulated miR-125b-5p and downregulated miR-218-5p, we also observed the upregulation of miR-99a-5p, miR-100-5p, and miR-125b-2-3p and the downregulation of miR-551b-3p, miR-551b-5p, and miR-383-5p in both TOP-resistant cell lines. More importantly, all these miRNAs, with the exception of miR-383-5p, were also changed in the A2780DR1 cell line. Previously, we observed the increased expression of miR-99a-5p in two A2780 CIS-resistant ovarian cancer cell lines [51]. In serum from ovarian cancer patients, the level of miR-99a-5p was also upregulated and correlated with cancer cell invasion in vitro [62]. This indicates that miR-99a-5p upregulation can be related to EOC invasion and drug resistance. Here, we observed the overexpression of miR-100-5p in two TOP-resistant cell lines and in the A2780DR1 cell line. Previously, overexpression of this miR was also observed in PAC-resistant cell lines [51,63] and ovarian cancer tissue [64]. Thus, this is the next miRNA that seems to be associated with drug-resistance development and ovarian cancer progression. We did not find any data concerning the role of miR-125b-2-3p in TOP or DOX resistance and in ovarian cancer. However, in contrast to our study, the expression of miR-125b-2-3p was rather decreased in other types of cancer. In colorectal cancer, tissue and cell line expression of this miR was decreased, and overexpression increased sensitivity to drugs [65]. In a similar way, miR-125b-2-3p was significantly lower in HCC than in non-tumor tissue [66]. The level of miR-125b-2-3p was also downregulated in head and neck squamous cell carcinoma [67]. The contrasting results suggest a cancer-type-dependent role of miR-125b-2-3p expression. In both TOP-resistant cell lines and in the A2780DR1 cell line, we also observed the downregulation of both strands of miR-551b (miR-551b-3p and miR-551b-5p). In a similar way, we previously observed the downregulation of these miRs in two A2780 PAC-resistant cell lines [51], which suggests a similar mechanism of drug resistance between PAC-, TOP-, and possibly DOX-resistant cell lines. Reviewing the literature data, we found that the decreased expression of miR-551b-5p was noted in breast cancer cell lines and tissue, and it was an indicator of patients’ poor overall survival (OS) [68]. On the other hand, the expression of miR-551b-3p was significantly decreased in gastric cancer and correlated with degree of differentiation, TNM stage, and lymph node metastasis [69]. The downregulation of miR-551b-3p was also observed in cholangiocarcinoma samples and cell lines, and it was linked to increased cell proliferation [70]. Other researchers observed the downregulation of miR-551b-3p and miR-551b-5p during tumor progression, and so did we in drug-resistant cell lines. As tumors progression is usually related to higher drug resistance, these miRNAs may be involved in both processes. Both TOP-resistant cell lines and the A2780DR1 cell line exhibited downregulation of miR-383-5p. It has been suggested that the downregulation of this miR can be related to PAC resistance, as the upregulation of this miRNA enhanced the sensitivity of OC cell lines to PAC [71], and we observed its downregulation in the A2780 PAC-resistant cell line [51]. We did not find any data concerning the role of miR-383-5p in TOP resistance. However, knockdown of miR-383 increased the resistance of hepatocellular carcinoma (HCC) to DOX [72], and we observed its downregulation in the DOX-resistant cell line as well. miR-383-5p is also considered as a potential tumor suppressor gene in ovarian cancer. In a study of HAN et al., miR-383 expression was significantly downregulated in OC tissues, and OC cell lines enhanced cell proliferation and inhibited apoptosis [73]. In another ovarian cancer study, strong downregulation of miR-383 was observed in high-grade serous OC (HGSC) and clear cell OC (CCC) compared with ovarian surface epithelium (OSE) [74]. Thus, downregulation of miR-383 seems to be associated with drug resistance and disease progression, not only in ovarian cancer but also in other cancers.

Next, we analyzed other miRNAs, the expression of which was significantly altered in tested drug-resistant cell lines; however, the pattern of their expression differed between cell lines resistant to the same drug. The highest expression was observed for miR-217 in the A2780DR1 cell line (over 200-fold), suggesting its significant role in resistance to DOX. In contrast to our results, a study on gastric cancer indicated that inhibition of miR-217 correlated with higher resistance to DOX and PAC [75]. Similarly, in acute myeloid leukemia (AML), patients exhibited decreased levels of miR-217, and overexpression of that miRNA increased their sensitivity to DOX [76]. In EOC tissue, miR-217 expression was downregulated and inversely correlated with advanced FIGO stage, high histological grading, and lymph node metastasis [77]. In the context of all these results, the role of miR-217 in our model requires further investigation. In the A2780DR1 cell line we also observed strong upregulation of miR-216a-5p and miR-216b-5p. We did not find any data about the role of these miRs in DOX resistance. However, the overexpression of miR-216b-5p in PAC-resistant ovarian cancer cell lines stimulated PAC sensitivity. Serous epithelial ovarian cancer patients with lower expression of miR-216b-5p also had shorter OS and PFS [78]. The tumor suppressive role of miR-216a-5p, achieved by inhibition of cell proliferation, was described in pancreatic, breast, and colorectal cancer, among others [79,80,81]. In summary, the tumor suppressive role and inhibition of drug resistance by these miRNAs seem to contrast our results. Previously, we also observed very strong overexpression of miR-205-5p in the PAC-resistant cell line. Here, we observed it upregulated in the A2780TR1 cell line; however, we did not find any data concerning the role of this miRNA in TOP resistance. In a study on patients, upregulation of miR-205-5p was observed in HGSC tumors in comparison to the ovarian surface epithelium [74]. The plasma level of miR-205-5p was upregulated in ovarian cancer patients, and even a higher level was observed in poorly differentiated tumors in compression to well/moderate differentiated ones [82]. It suggests that the overexpression of miR-205-5p can be related to ovarian cancer progression and drug resistance. In the TOP-resistant cell line, we also observed the upregulation of miR-379-5p; however, we did not find any data linking the expression of this miR in TOP resistance or ovarian cancer. In contrast to our results, the decreased expression of miR-379-5p was observed in NSCLC in comparison to the control [83]. In breast cancer, the level of miR-379 expression was significantly decreased compared to normal breast tissues and was also found to decrease significantly with increasing tumor stage [84].

Summarizing this part of the research, we can conclude that changes in expression of miR-935, miR-218-5p, miR-99a-5p, miR-100-5p, miR-551b-5p, miR-551b-3p, miR-383, and miR-205-5p were in accordance with the literature data, although we did not find any conformation of their expression in TOP-resistant cell lines. This is probably due to the limited availability of and scarce research on TOP-resistant cell lines. On the other hand, the expression of miR-125b-5p, miR-125b-2-3p, miR-217, miR-216a-5p, miR-216b-5p, and miR-379-5p was contrasting to the results from the literature. Our results suggest the potential role of miRNA in drug-resistance development. Especially, miRNAs down/upregulated in two cell lines resistant to DOX or TOP seem to be important. We also observed differences in the expression of particular miRNAs between two cell lines resistant to the same drug, which indicates a complication of the drug-resistance mechanism development, which we also observed previously for the protein-encoding genes [24,53,54].

The interpretation of changes in miRNA expression is much more difficult than of protein-coding genes. For example, the expression of drug transporters, such as P-gp or BCRP, is nearly always involved in resistance to DOX or TOP, respectively [85]. The role of miRNA expression is more complex, as one miRNA may regulate the expression of many genes, and one mRNA can be a target of different miRs. The other difficulty in interpretation of miRNAs expression is that in contrast to protein coding genes, miRNAs are not directly involved in drug resistance, but instead they regulate gene expression not only directly, by regulating drug-resistant genes, but also indirectly, by modulation of the signaling pathways controlling gene expression. To better explain the significance of the investigated miRNA to drug resistance, the second part of our research was focused on a reverse correlation between the expression of miRNAs and their target genes. We analyzed mainly genes previously described by us in the context of drug resistance [24,53,54].

The role of ECM molecule expression in development of drug resistance in cancers has been studied for many years [28,86,87]. However, ECM molecule expression is not limited to tumors, but was also observed in drug-resistant breast [88] and ovarian cancer cell lines [89]. Previously, we observed changes in the expression of many ECM genes in the investigated cell lines [54]. Here, we noticed that the expression of some of these ECM genes is regulated by miRNAs. Among others, we observed that overexpression of *CDH2* in the A2780DR1 cell line is associated with the downregulation of miR-218-5p. Similar observations were made by our team in CIS-resistant cell lines [51]. The regulation of CDH2 by miR-218-5p has been also reported in lung adenocarcinoma [90]. In the A2780TR2 cell line, we observed that increased expression of two other ECM molecules, *MMP1* and *COL1A2*, correlated with the downregulation of let-7i-5p. The regulation of MMP1 expression by let-7i-5p was confirmed experimentally in human fibroblasts [91]. Furthermore, the regulation of COL1A2 expression by let-7i-5p was reported in cultured cardiac fibroblasts [92]. Interestingly, although we observed a similar increase of *COL1A2* in both TOP-resistant cell lines [31], 7i-5p was a likely regulating factor in only one of them.

Reversible phosphorylation/dephosphorylation is one of several important cellular signaling pathways regulating gene expression, cellular metabolism, and rate of cell proliferation [93]. Cancer cells usually present higher levels of protein phosphorylation [94], and increased protein phosphorylation can be associated with resistance to drugs [26,27,95]. Increased protein phosphorylation can be associated with the increased expression of protein kinases and the decreased expression of protein phosphatases, as we observed previously [17,24,26,27]. Here, we observed that the expression of both types of enzymes can be regulated by miRNA. From our point of view, the most important phosphatase was PTPRK, as we observed its downregulation in 17 drug-resistant ovarian cancer cell lines [26], and others observed that downregulation of PTPRK correlates with poor prognosis in breast cancer [96] and worse response to chemotherapy in nasal-type NK/T-cell lymphoma (NKTCL) [97]. We observed that the expression of *PTPRK* is regulated by miRNAs in A2780DR2, A2780TR1, and A2780TR2 cell lines. In detail, we noticed an inverse correlation between *PTPRK* and miR-935 in A2780DR2 cell line, and it was the first such observation made. In the A2780TR1 cell line, *PTPRK* expression was regulated by miR-409-3p. Similar regulation was previously noted by us in one of the CIS-resistant cell lines [51]. However, in the second TOP-resistant cell line, *PTPRK* expression was regulated by miR-363-3p. Thus, the expression of the same gene, even in twin cell lines resistant to the same drug, was regulated by different miRNAs. In a similar way, the regulation of another phosphatase, *PTPRZ1*, by different miRs in both TOP-resistant cell lines was observed in A2780TR1 by miR-431-5p and in A2780TR2 by miR-20b-3p. Regulation of *PTPRZ*1 by miR-431-5p was also previously observed by our team in the CIS-resistant cell line [51]. In the investigated cell lines, we also noticed the increased expression of kinases, especially EPHA7 [25]. EPHA7 encodes receptor tyrosine kinase, which activates the ERK pathway [98]. The increased expression of EPHA7 was observed in different cancers, among others in hepatocellular carcinoma, and was associated with tumor progression, invasion, and metastasis [99]. Here we observed the upregulation of *EPHA7* expression by miR-218-5p. A similar pattern of regulation was observed by us previously in both A2780-PAC-resistant cell lines [51]. *EPHA7* was also a putative target of miR-218-5p in a human airway epithelium study [100].

ROBO2, a transmembrane receptor, is a putative tumor suppressor gene with decreased expression in different cancers. Downregulation of ROBO2 expression was observed in the poorly differentiated SKOV-3 ovarian cancer cell line in comparison to the more differentiated PEO-14 ovarian cancer cell line [101]. We observed the downregulation of *ROBO2* in the TOP-resistant A2780TR2 cell line, and previously we also observed the downregulation of *ROBO2* in two CIS-resistant ovarian cancer cell lines [51], which suggests that the role of ROBO2 downregulation is in drug-resistance development. In contrast to our previous study, here we noted that *ROBO2* downregulation correlates with increased expression of miR-363-3p. This kind of regulation was probably observed for the first time. *SAMD4A* is a translational repressor of genes encoding SRE-containing messenger proteins [102]. Previously, we observed its upregulation in three TOP-resistant cell lines [17,24] and two PAC-resistant cell lines [24], suggesting its role in drug-resistance development. Here we noted that in both TOP-resistant cell lines, the increased expression of *SAMD4* correlates with the decreased expression of miR-551b-5p. An identical pattern of regulation was also observed by us previously in both PAC-resistant cell lines [51], which suggests that this is a universal mechanism of *SAMD4* gene expression regulation in our model.

In both TOP-resistant cell lines, we observed the overexpression of *MARCKS*. As we did not describe it previously, this time we focused in detail on the physiological and pathological roles of this protein. MARCKS is the main target of protein kinase C. In the unphosphorylated form, it is bound to the plasma membrane, and after phosphorylation shuttling to the cytoplasm, it regulates different cellular processes, such as cytoskeletal reorganization, membrane trafficking, cell secretion, inflammatory response, cell migration, and mitosis [103,104]. It has been reported that MARCKS is associated with the development and progression of hematological cancers [103]. The role of MARCKS in tumorigenesis, metastasis, and resistance to anti-cancer therapies of other cancers has also been described [104]. In a study concerning triple-negative breast cancer (TNBC), the direct role of MARCKS in resistance to PAC has been reported [105]. The overexpression of MARCKS has also been reported in several drug-resistant human melanoma cell lines (HMCLs) and in drug-resistant primary multiple melanoma (MM) [106]. Furthermore, knockdown of the *MARCKS* gene or inhibition of MARCKS phosphorylation increased sensitivity to anti-melanoma drugs [106]. The role of MARCKS in the drug resistance of ovarian cancer has never been described. Previously, we observed the overexpression of *MARCKS* in two PAC-resistant ovarian cancer cell lines [51], and the role of MARCKS in PAC resistance has been reported in breast cancer [105]. Here we observed overexpression in two TOP-resistant ovarian cancer cell lines. We did not find any data about the role of this protein in TOP resistance, probably due to the limited number of TOP-resistant cell lines used in research. Interestingly, in all four drug-resistant cell lines, we observed that *MARCKS* overexpression correlates with the downregulation of miR-218-5p and miR-551b-5p, suggesting similar regulation of this protein expression in all of the cell lines. Curiously, downregulation of miR-218-5p was associated with ovarian cancer progression [61], and downregulation of miR-551b-5p was linked with the progression of breast cancer.

The most important protein involved in TOP resistance is the BCRP protein encoded by the *ABCG2* gene [14,85]. We also observed the overexpression of this protein in all investigated TOP-resistant ovarian cancer cell lines [16,17,18]. Here we noted that in one of two A2780 TOP-resistant cell lines, the overexpression of the *ABCG2* gene correlated with the downregulation of miR-212-3p. A similar observation was made in leukemia cells, where inhibition of miR-212 led to increased expression of *ABCG2* and imanitib resistance [107]. The regulation of ABCG2 expression by miR-212-3p was also observed in clear cell renal cell carcinoma [108]. Thus, our observation was confirmed by the literature data. However, another question was raised: Why is the *ABCG2* gene regulated by miR-212-3p only in one of the twin TOP-resistant cell lines?

In summary, we detected many miRNA–target pairs. Some of these pairs were described in the literature, while others were described by us for the first time. Interestingly, even in the twin cell lines resistant to the same drug, the regulation of one target was carried out by different miRNAs, which indicates how complicated the mechanisms are that are involved in miRNA-regulated gene expression.

Although many changes of miRNA expression levels in DOX- and TOP-resistant ovarian cancer cell lines were described in this study, and some of them were identified as potential targets, we must not forget that these results have their limitations. To further explain the role of investigated miRNAs in drug resistance and regulation of gene and protein expression, a functional study is needed. This would include the silencing and overexpression of selected miRNAs performed in vitro. These experiments will be performed as a continuation of this research to explain the exact function of the investigated miRNAs. Additionally, the study on ovarian cancer tissue from chemotherapy-sensitive and -resistant patients should be performed to confirm the significance of the investigated miRNAs in drug resistance. If we are successful, it could confirm that miRNAs are the potential new targets for general or personalized therapy in ovarian cancer that can limit ovarian cancer progression and/or increase the efficiency of chemotherapy.

## 4. Materials and Methods

### 4.1. Reagents

Doxorubicin and topotecan were acquired from Sigma (St. Louis, MO, USA). RPMI-1640 medium, fetal bovine serum, penicillin, streptomycin, amphotericin B (25 μg/mL), and L-glutamine were also bought from Sigma. QIazol Lysys Reagent, miRNeasy Mini Kit, and RNeasy MinElute Cleanup Kit were purchased from Qiagen (Hilden, Germany). GeneChipTM miRNA 4.1 Array Strip, FlashTagTM Biotin HSR RNA Labeling Kits, GeneAtlasTM Hybridization, Wash, and Stain Kit for miRNA Arrays were acquired from Afymetrix (Santa Clara, CA, USA).

### 4.2. Cell Lines and Cell Culture

The human ovarian carcinoma A2780 cell line was purchased from ATCC. A2780 sublines that were resistant to DOX (A2780DR1, A2780DR2 (A2780 doxorubicin-resistant)), and TOP (A2780TR1 and A2780TR2 (A2780 topotecan-resistant)) were obtained by exposing A2780 cells to the relevant drugs at gradually increasing concentrations. The final concentrations used to select the resistant cells were 100 ng/mL of DOX for both A2780DR1 and A2780DR2 cell lines, and 24 ng/mL of TOP for both A2780TR1 and A2780TR2 cell lines, as described previously [16]. They were two-fold greater than the concentrations in the plasma 2 h after intravenous administration [109]. According to parental drug-sensitive cell lines, the increase in resistance was as follows: 58-fold for A2780DR1 vs. A2780, 73-fold for A2780DR2 vs. A2780, 60-fold for A2780TR1 vs. A2780, and 48-fold for A2780TR2 vs. A2780, as we described previously [16]. The cell lines were cultured as a monolayer in a complete medium (MEM medium with the addition of 10% (*v*/*v*) fetal bovine serum, 2 pM L-glutamine, penicillin (100 U/mL), streptomycin (100 U/mL), and amphotericin B (25 µg/mL)) at 37 °C in a 5% CO2 atmosphere.

### 4.3. miRNA Isolation

miRNA was isolated with a Qiagen reagent kit as instructed in the producer’s protocol. The concentration and quality of isolated miRNA were quantified by probing absorbance at 260 nm and 280 nm. The 260/280 nm ratio was used to estimate the degree of protein contamination. In all of the samples, the 260/280 nm ratio oscillated between 1.8 and 2.0. From each of the studied groups, four samples were selected for the miRNA expression profiling by microarray (N/group = 4).

### 4.4. Microarray Preparation, Hybridization, and Scanning

The miRNA expression profiling was carried out using Affymetrix platform-based microarrays with GeneChip™ miRNA 4.1 Array Strip (ThermoFisher Scientific, Waltham, MA, USA). The technical procedure that was followed was previously described in detail elsewhere [110,111]. Each microarray was designed following the miRBase Release 20 database, which included complementary probes for 2578 human mature miRNA, 1996 human snoRNA, CDBox RNA, H/ACA Box RNA, scaRNA, and 2025 human pre-miRNA. miRNA was prepared for the hybridization step with the FlashTagTM Biotin HSR RNA Labeling Kit (ThermoFisher Scientific, Waltham, MA, USA). During this process, 150 ng of miRNA was subjected to the poly(A) tailing and biotin ligation in accordance with the producer’s protocol. Biotin-labeled miRNA were then hybridized to GeneChip™ miRNA 4.1 Array Strip (20 h, 48 °C). Afterwards, the microarrays were washed and stained following the technical protocol and using the Affymetrix GeneAtlas Fluidics Station (Affymetrix, Santa Clara, CA, USA). Next, the array strips were scanned using GeneAtlas System Imaging Station (Thermo Fisher Scientific, Waltham, MA, USA).

### 4.5. Microarray Analysis and miRNA Gene Screening

The preliminary analysis of scanned microarrays was performed using Affymetrix GeneAtlas Operating Software (Affymetrix, Santa Clara, CA, USA). The quality of the miRNA expression profile was verified with the default quality control criteria set in the software. BioConductor libraries from the statistical programming language “R” were used for further analyses. The raw CEL files obtained after microarray scanning were imported into the programming environment. The Robust Multiarray Average (RMA) normalization algorithm (implemented in the “Affy” library) was used for normalization, correction of the background, and the expression calculation [112]. To assign microarray quality and diagnose batch effects, we used the “array QualityMetrics” library. The biological annotation for human miRNA was taken from the pd.mirna 4.1 library, where tabulated biological descriptors were combined with a miRNA normalized dataset to obtain a full miRNA data table. Differential expression and statistical evaluation were calculated using the “limma” library based on the linear model for microarray data [113]. P-values were determined using an empirical Bayes moderated t-test with false discovery rate (FDR) correction for multiple tests. The selection criteria for significantly regulated miRNA expression were based on the variance between folds greater than absolute five and p-values after FDR correction (adj.p.val) <0.05. The results obtained were then visualized as scatter plots showing the total number of miRNAs up- and downregulated. Differently expressed miRNA were also presented as heat maps and tables. Raw data files were deposited into the Gene Expression Omnibus (GEO) repository at the National Center for Biotechnology Information (http://www.ncbi.nlm.nih.gov/geo/, accessed on 3 January 2022) under the GEO accession number GEO: GSE190245 (public on 3 January 2022).

### 4.6. miRNA-Target Gene Prediction

To identify the potential target genes for differently expressed miRNA, we used the SpidermiR library, where symbols of differentially expressed miRNAs were applied to find target genes in databases as follows: for predicted targets—DIANA, Miranda, PicTar, and TargetScan, and for experimentally validated targets—miRTAR and miRwalk [114]. The mRNA transcriptomic data from our previously published experiment were used to determine the actual expression values of the target genes [24,53,54]. The obtained fold change values for mRNA were combined with the target genes’ data table. For further analysis, we selected only the target genes of which the fold change was inversely correlated with the fold change value of the appropriate miRNA (cut-off criteria: fold ± 5, adjusted *p* value (adj.p.val.) <0.05). From the entire collection of miRNA–targets pairs, we then selected only those that were involved in cancer stem cell biology, drug resistance, and the extracellular matrix, applying the following GO terms keywords: “collagen-containing extracellular matrix”, “extracellular matrix”, “extracellular space”, “response to drug”, and “stem cell”. Interactions between miRNA and target genes in relation to selected GO terms were visualized with Cytoscape 3.7.2 (National Institute of General Medical Sciences, Bethesda, MD, USA) [115].

## 5. Conclusions

In summary, we identified miRNAs that may be specific to DOX or TOP resistance, as changes in their expression were present in both DOX- and/or TOP-resistant cell lines. Some of them were previously described as important factors in cancer progression and/or development of drug resistance. Abnormal expression of some of the other miRNAs was observed for the first time. We also noticed an inverse correlation between the expression of miRNAs and their targets. The role of some of these targets was previously described in the context of drug resistance and/or cancer progression. The interactions between new pairs should be investigated in detail. The significance of these interactions in the context of resistance to cytotoxic drugs needs further investigation.

## Figures and Tables

**Figure 1 ijms-23-05846-f001:**
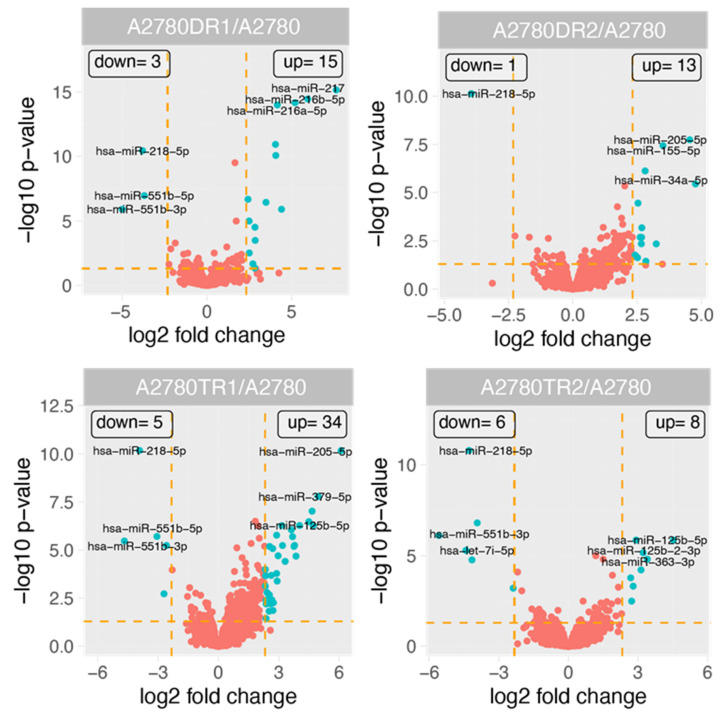
Volcano plots representing the miRNA with at least 5-fold upregulated or downregulated expression levels (green dots) in drug-resistant cell line, compared with A2780 cell line. Red dots show miRNAs below the established cut-off criteria (|fold|> 5, *p* < 0.05).

**Figure 2 ijms-23-05846-f002:**
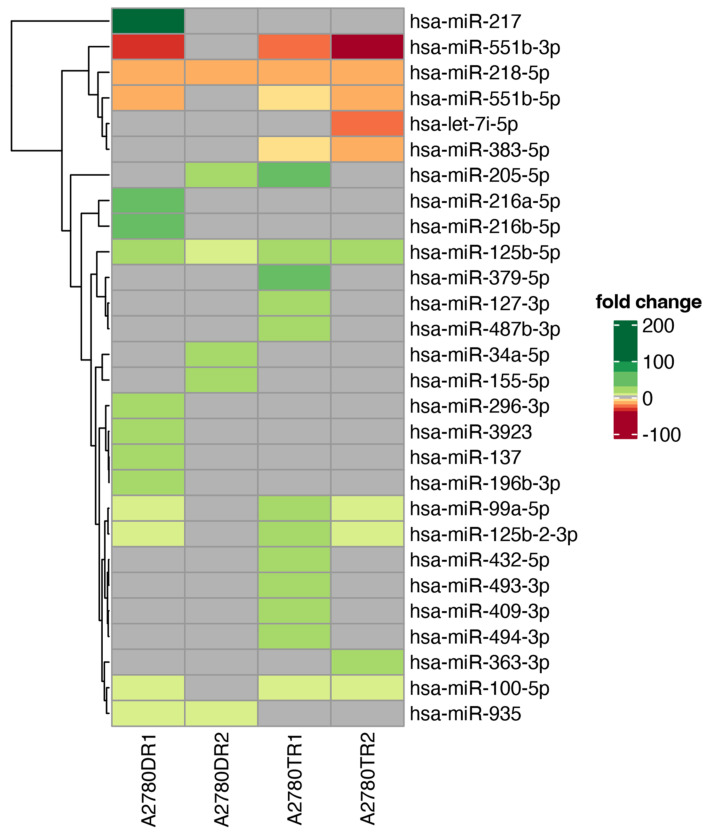
Heatmap with hierarchic clustering of miRNA expression ratios in drug-resistant sublines. Grey boxes correspond to genes below the cut-off criteria (|fold| < 5, *p* > 0.05). Differentially expressed genes were marked by color scale (up-regulated—light to dark green, down-regulated—orange to red). Expression change was calculated in relation to the A2780 cell line.

**Figure 3 ijms-23-05846-f003:**
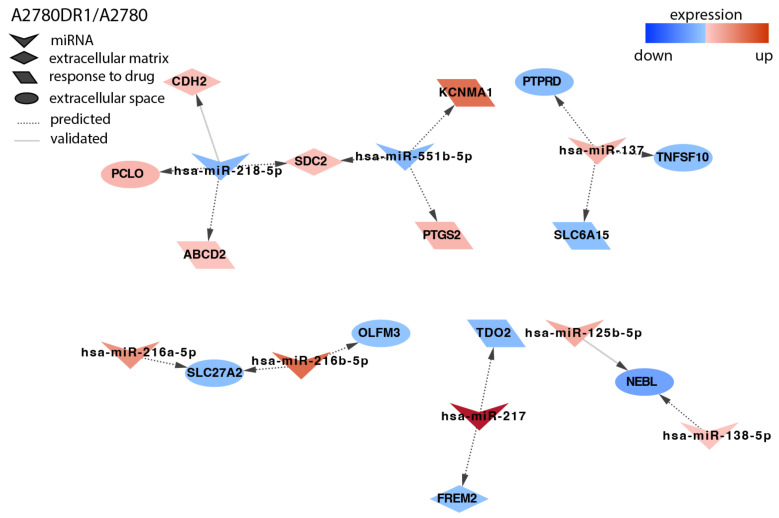
Regulation of selected target genes by miRNAs in the A2780DR1 cell line.

**Figure 4 ijms-23-05846-f004:**
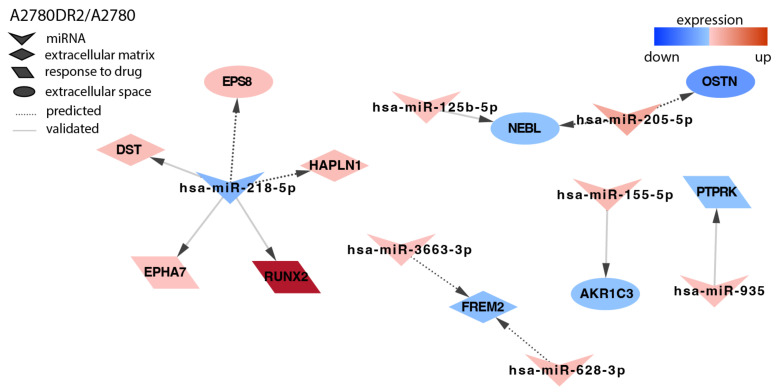
Regulation of selected target genes by miRNAs in the A2780DR2 cell line.

**Figure 5 ijms-23-05846-f005:**
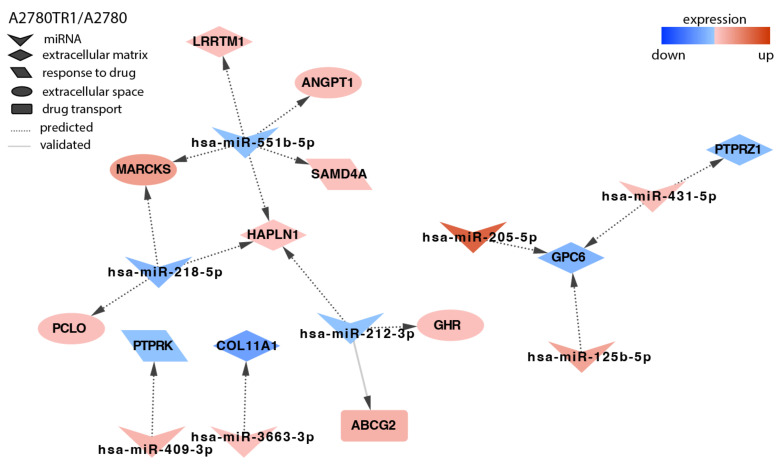
Regulation of selected target genes by miRNAs in the A2780TR1 cell line.

**Figure 6 ijms-23-05846-f006:**
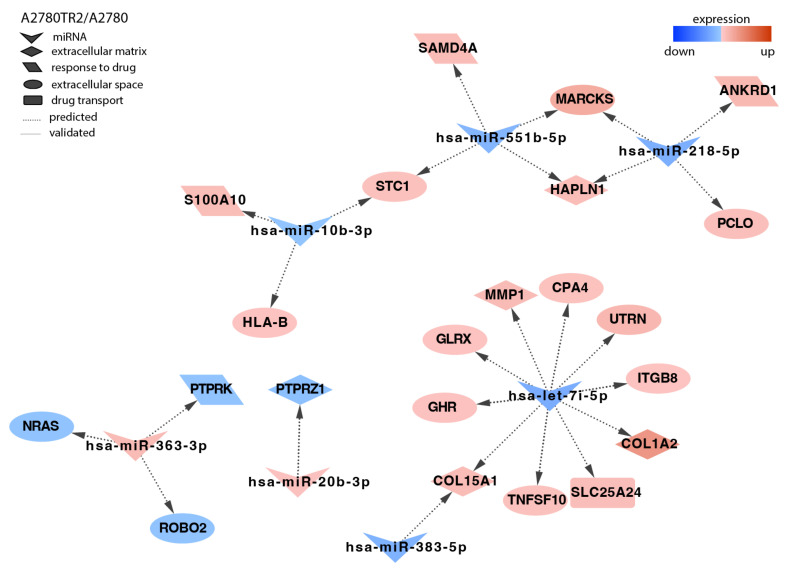
Regulation of selected target genes by miRNAs in the A2780TR2 cell line.

**Table 1 ijms-23-05846-t001:** List of the miRNA fold changes and false discovery rate (FDR) corrected *p*-values (adj.p.val.). Each comparison was performed in relation to A2780 cells. N.S: up/downregulation between 5 and −5, or insignificant alterations.

miRBase Accession Number	Gene Name	Fold Change (adj.p.val)			
		A2780DR1	A2780DR2	A2780TR1	A2780TR2
MIMAT0000097	hsa-miR-99a-5p	7.23575965	N.S	13.4763932	6.94690251
MIMAT0000098	hsa-miR-100-5p	5.64229368	N.S	5.82903562	7.58322275
MIMAT0000252	hsa-miR-7-5p	N.S	N.S	5.76802822	N.S
MIMAT0000255	hsa-miR-34a-5p	N.S	27.3069918	N.S	N.S
MIMAT0000266	hsa-miR-205-5p	N.S	23.4285218	68.1826341	N.S
MIMAT0000269	hsa-miR-212-3p	N.S	NS	−5.98258494	N.S
MIMAT0000273	hsa-miR-216a-5p	37.0885659	N.S	N.S	N.S
MIMAT0004959	hsa-miR-216b-5p	62.2848826	N.S	N.S	N.S
MIMAT0000274	hsa-miR-217	200.01382	N.S	N.S	N.S
MIMAT0000275	hsa-miR-218-5p	−13.9723251	−15.4598435	−15.1300271	−19.3883641
MIMAT0000415	hsa-let-7i-5p	N.S	N.S	N.S	−21.2652763
MIMAT0000423	hsa-miR-125b-5p	21.2850391	6.14958848	26.0504456	23.089459
MIMAT0004603	hsa-miR-125b-2-3p	7.17389557	N.S	12.3020071	9.34279559
MIMAT0003233	hsa-miR-551b-3p	−32.0993258	N.S	−25.4904329	−47.6292686
MIMAT0004794	hsa-miR-551b-5p	−12.9832503	N.S	−8.30875013	−15.1855837
MIMAT0000429	hsa-miR-137	16.8235232	N.S	N.S	N.S
MIMAT0000430	hsa-miR-138-5p	5.67137662	N.S	N.S	N.S
MIMAT0000446	hsa-miR-127-3p	N.S	N.S	22.4289639	N.S
MIMAT0000447	hsa-miR-134-5p	N.S	N.S	9.93993656	N.S
MIMAT0000646	hsa-miR-155-5p	N.S	11.3736149	N.S	N.S
MIMAT0000707	hsa-miR-363-3p	N.S	N.S	N.S	10.6015891
MIMAT0000722	hsa-miR-370-3p	N.S	N.S	6.47195247	N.S
MIMAT0000733	hsa-miR-379-5p	N.S	N.S	30.945542	N.S
MIMAT0000737	hsa-miR-382-5p	N.S	N.S	9.07152577	N.S
MIMAT0000738	hsa-miR-383-5p	N.S	N.S	−6.51726481	−17.7612762
MIMAT0001625	hsa-miR-431-5p	N.S	N.S	8.87926645	N.S
MIMAT0001639	hsa-miR-409-3p	N.S	N.S	14.4397796	N.S
MIMAT0002178	hsa-miR-487a-3p	N.S	N.S	7.4124047	N.S
MIMAT0002814	hsa-miR-432-5p	N.S	N.S	13.744849	N.S
MIMAT0002816	hsa-miR-494-3p	N.S	N.S	16.3141937	N.S
MIMAT0003161	hsa-miR-493-3p	N.S	N.S	12.8494251	N.S
MIMAT0003180	hsa-miR-487b-3p	N.S	N.S	25.1135622	N.S
MIMAT0003239	hsa-miR-574-3p	N.S	5.79962256	N.S	N.S
MIMAT0003297	hsa-miR-628-3p	N.S	9.47532594	N.S	N.S
MIMAT0004556	hsa-miR-10b-3p	−2.842098067	−4.763634832	−2.347192792	−5.183265974
MIMAT0004679	hsa-miR-296-3p	11.258982	N.S	N.S	N.S
MIMAT0004752	hsa-miR-20b-3p	N.S	N.S	N.S	6.45001658
MIMAT0004776	hsa-miR-505-5p	6.59829128	N.S	N.S	NS
MIMAT0004951	hsa-miR-887-3p	NS	6.42842883	N.S	8.77616954
MIMAT0004978	hsa-miR-935	7.10687862	7.1679309	N.S	N.S
MIMAT0005865	hsa-miR-1202	N.S	N.S	7.40030689	N.S
MIMAT0009201	hsa-miR-196b-3p	16.6120623	N.S	N.S	N.S
MIMAT0010251	hsa-miR-449c-5p	N.S	N.S	1.87888757	6.63203316
MIMAT0015070	hsa-miR-3188	N.S	N.S	6.29125598	N.S
MIMAT0015079	hsa-miR-3195	N.S	7.03029052	N.S	N.S
MIMAT0018085	hsa-miR-3663-3p	1.956795219	6.39351738	7.544158456	4.146984658
MIMAT0018198	hsa-miR-3923	17.9229767	N.S	N.S	N.S
MIMAT0018968	hsa-miR-4449	N.S	5.72863128	5.72224812	N.S
MIMAT0019691	hsa-miR-4634	N.S	N.S	5.06702643	N.S
MIMAT0019979	hsa-miR-4800-3p	N.S	N.S	5.32867026	N.S
MIMAT0022838	hsa-miR-1185-1-3p	N.S	N.S	6.15646443	N.S
MIMAT0023693	hsa-miR-6068	N.S	N.S	5.15633061	N.S
MIMAT0023700	hsa-miR-6075	N.S	N.S	5.02749704	N.S
MIMAT0023705	hsa-miR-6080	N.S	5.36404532	6.00292053	N.S
MIMAT0027480	hsa-miR-6790-5p	N.S	N.S	6.63324861	N.S
MIMAT0027678	hsa-miR-6889-5p	N.S	N.S	5.1618654	N.S
MIMAT0028121	hsa-miR-7112-5p	N.S	6.3493949	5.16702158	N.S
MIMAT0028213	hsa-miR-7151-3p	5.39257884	N.S	N.S	N.S
MIMAT0030420	hsa-miR-7845-5p	N.S	N.S	5.02021002	N.S
MIMAT0030990	hsa-miR-8063	N.S	N.S	6.53874406	N.S
MIMAT0030991	hsa-miR-8064	N.S	N.S	7.80350724	N.S

**Table 2 ijms-23-05846-t002:** List of the miRNA fold changes and false discovery rate (FDR)-corrected *p* values (adj.p.val.). Each comparison was performed in relation to A2780 cells. N.S: up/downregulation between 5 and −5, or insignificant alterations.

miRBase Accession Number	Gene Name	Fold Change (adj.p.val)			
		A2780DR1	A2780DR2	A2780TR1	A2780TR2
MIMAT0000097	hsa-miR-99a-5p	7.23575965	N.S	13.4763932	6.94690251
MIMAT0000098	hsa-miR-100-5p	5.64229368	N.S	5.82903562	7.58322275
MIMAT0000255	hsa-miR-34a-5p	N.S	27.3069918	N.S	N.S
MIMAT0000266	hsa-miR-205-5p	N.S	23.4285218	68.1826341	N.S
MIMAT0000273	hsa-miR-216a-5p	37.0885659	N.S	N.S	N.S
MIMAT0004959	hsa-miR-216b-5p	62.2848826	N.S	N.S	N.S
MIMAT0000274	hsa-miR-217	200.01382	N.S	N.S	N.S
MIMAT0000275	hsa-miR-218-5p	−13.9723251	−15.4598435	−15.1300271	−19.3883641
MIMAT0000415	hsa-let-7i-5p	N.S	N.S	N.S	−21.2652763
MIMAT0000423	hsa-miR-125b-5p	21.2850391	6.14958848	26.0504456	23.089459
MIMAT0004603	hsa-miR-125b-2-3p	7.17389557	N.S	12.3020071	9.34279559
MIMAT0003233	hsa-miR-551b-3p	−32.0993258	N.S	−25.4904329	−47.6292686
MIMAT0004794	hsa-miR-551b-5p	−12.9832503	N.S	−8.30875013	−15.1855837
MIMAT0000429	hsa-miR-137	16.8235232	N.S	N.S	N.S
MIMAT0000446	hsa-miR-127-3p	N.S	N.S	22.4289639	N.S
MIMAT0000646	hsa-miR-155-5p	N.S	11.3736149	N.S	N.S
MIMAT0000707	hsa-miR-363-3p	N.S	N.S	N.S	10.6015891
MIMAT0000733	hsa-miR-379-5p	N.S	N.S	30.945542	N.S
MIMAT0000738	hsa-miR-383-5p	N.S	N.S	−6.51726481	−17.7612762
MIMAT0001639	hsa-miR-409-3p	N.S	N.S	14.4397796	N.S
MIMAT0002814	hsa-miR-432-5p	N.S	N.S	13.744849	N.S
MIMAT0002816	hsa-miR-494-3p	N.S	N.S	16.3141937	N.S
MIMAT0003161	hsa-miR-493-3p	N.S	N.S	12.8494251	N.S
MIMAT0003180	hsa-miR-487b-3p	N.S	N.S	25.1135622	N.S
MIMAT0004679	hsa-miR-296-3p	11.258982	N.S	N.S	N.S
MIMAT0004978	hsa-miR-935	7.10687862	7.1679309	N.S	N.S
MIMAT0009201	hsa-miR-196b-3p	16.6120623	N.S	N.S	N.S
MIMAT0018198	hsa-miR-3923	17.9229767	N.S	N.S	N.S

## Data Availability

Not applicable.

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
