# Peer review of "The Profile of MicroRNA Expression and Potential Role in the Regulation of Drug-Resistant Genes in Doxorubicin and Topotecan Resistant Ovarian Cancer Cell Lines"

_ijms, 2022, doi:10.3390/ijms23105846_

Round 1
Reviewer 1 Report
Overall an interesting and well-written manuscript on miRNA expression profiles and potential drug-resistant genes in ovarian cancer cell lines.
To assist readability and emphasis the significance of the findings, authors are encouraged to consider the following comments:
- Revise grammar on lines 41, 46, 82 and 88.
- Please check the meaning on 102 given there are over 2000 miRNA reported.
- Section 2.3, it is recommended to state which miRNA you are referring to (I.e which "5 significantly changed") or reference which table to refer to.
- Table 1, please define N.S
- Fig 2, please provide more information/significance of the data shown in the ledger
- Methods section, how was technical error controlled? I.e replicates, repeat testing?
- Discussion section, please provide further comment on the limitations to the work/recommendations for further studies. I.e functional studies required?
- Discussion section, please provide/clarify what functional studies are required to support the findings or link to literature that provides these justifications for the significance of the data and its impact on Ovarian cancer.
Author Response
Response to Reviewer 1 Comments
Thank you for reading the manuscript and for all of your comments and questions. Please find our responses below. I hope that they will provide you with adequate answers and explanations.
1. Revise grammar on lines 41, 46, 82 and 88.
Response 1. The grammar has been revised.
2. Please check the meaning on 102 given there are over 2000 miRNA reported.
Response 2. The sentence has been rewritten and the cited paper has been changed.
3. Section 2.3, it is recommended to state which miRNA you are referring to (I.e which "5 significantly changed") or reference which table to refer to.
Response 3. We changed the sentences and we made references to to the tables’ descriptions so now it should be clearer.
4. Table 1, please define N.S.
Response 4. The N.S. has been defined in Table 1 and Table 2.
5. Fig 2, please provide more information/significance of the data shown in the ledger.
Response 5. A relevant description was added to the figure 2 caption.
6. Methods section, how was technical error controlled? I.e replicates, repeat testing?
Response 6. The experiment was carried out in four repetitions – this was included in the materials and methods. Usually, technical repetitions of the same samples are not used in microarray experiments. Such a procedure would significantly increase the costs of experiment. Moreover, it generates much less bias than biological replicates. Quality and batch effect were verified using the "arrayQualityMetrics" library. The corresponding sentence was added to the main text.
7. Discussion section, please provide further comment on the limitations to the work/recommendations for further studies. I.e functional studies required?
8. Discussion section, please provide/clarify what functional studies are required to support the findings or link to literature that provides these justifications for the significance of the data and its impact on Ovarian cancer.
Response 7 and 8. At the end of the discussion chapter we added an additional paragraph describing the limitation of the study and its impact on the ovarian cancer treatment.

Reviewer 2 Report
No comments/suggestions.
Author Response
Response to Reviewer 2 Comments
We are grateful for your time and effort put into reviewing our manuscript. We hope that it made a good read.
